# *ZmSMR10* Increases the Level of Endoreplication of Plants through Its Interactions with *ZmPCNA2* and *ZmCSN5B*

**DOI:** 10.3390/ijms25063356

**Published:** 2024-03-15

**Authors:** Lulu Bao, Jihao Si, Mingming Zhai, Na Liu, Haoran Qu, Christian Capulong, Jinyuan Li, Qianqian Liu, Yilin Liu, Chenggang Huang, Maoxi Zhang, Zhengxiong Ao, Aojun Yang, Chao Qin, Dongwei Guo

**Affiliations:** 1Key Laboratory of Biology and Genetic Improvement of Maize in Arid Area of Northwest Region, Ministry of Agriculture, College of Agronomy, Northwest A&F University, Yangling 712100, China; baolulu03092022@163.com (L.B.); sijihao3487896@163.com (J.S.); zmm0810@nwafu.edu.cn (M.Z.); m18892088387@163.com (N.L.); 13936898695@163.com (H.Q.); christiancapulong05@nwafu.edu.cn (C.C.); lijy9906@163.com (J.L.); quan001221@163.com (Q.L.); greymajo@nwafu.edu.cn (Y.L.); cghuang8395@163.com (C.H.); zmx13505478987@163.com (M.Z.); aozhengxiong@163.com (Z.A.); 15616958148@163.com (A.Y.); qc249@nwafu.edu.cn (C.Q.); 2State Key Laboratory of Stress Biology for Arid Areas, Northwest A&F University, Yangling 712100, China

**Keywords:** endoreplication, maize, *ZmSMR10*, growth, development, interactions, stress

## Abstract

As a plant-specific endoreplication regulator, the SIAMESE-RELATED (*SMR*) family (a cyclin-dependent kinase inhibitor) plays an important role in plant growth and development and resistance to stress. Although the genes of the maize (*Zea mays*) *SMR* family have been studied extensively, the *ZmSMR10* (Zm00001eb231280) gene has not been reported. In this study, the function of this gene was characterized by overexpression and silencing. Compared with the control, the transgenic plants exhibited the phenotypes of early maturation, dwarfing, and drought resistance. Expression of the protein in prokaryotes demonstrates that ZmSMR10 is a small protein, and the results of subcellular localization suggest that it travels functionally in the nucleus. Unlike ZmSMR4, yeast two-hybrid experiments demonstrated that ZmSMR10 does not interact strongly with with some cell cycle protein-dependent protein kinase (CDK) family members ZmCDKA;1/ZmCDKA;3/ZmCDKB1;1. Instead, it interacts strongly with ZmPCNA2 and ZmCSN5B. Based on these results, we concluded that ZmSMR10 is involved in the regulation of endoreplication through the interaction of ZmPCNA2 and ZmCSN5B. These findings provide a theoretical basis to understand the mechanism of the regulation of endoreplication and improve the yield of maize through the use of molecular techniques.

## 1. Introduction

Cell-cycle regulation is an important mechanism of the adaptation of plants to internal and external stimuli [1]. Endoreplication, as one of the important models, is often defined as the phenomenon in which the cells do not divide and only replicate DNA internally; this results in the exponential growth of DNA in the cell and the formation of polyploidy [2]. The most critical cell-cycle regulatory proteins in this process include cyclin-dependent protein kinase (CDK) and cyclin (CYC). Different CDKs and CYC proteins have been found to bind to form various complexes that regulate other downstream target genes, such as pRb and E2F [3]. Studies have delineated two reasons for the transition of cells from mitosis to endoreplication, including inhibition of the M phase and promotion of the S phase. This process may be achieved by the upregulation of specific CDKs or CYC inhibitors, including p53, p21, and cyclin-dependent kinase inhibitors (CKIs) [4].

Endoreplication usually occurs in some highly differentiated organs that are highly metabolically active and plays an important role in cell differentiation, metabolic activity, and increase in the cell volume in plants [5]. In tomato (*Solanum lycopersicum*), the high level of endoreplication causes the enlargement of pericarp cells, which promotes fruit growth and development; the weight of fruit positively correlates with the size of pericarp cells [6]. In *Arabidopsis thaliana*, the degree of replication in the nucleus is crucial to the number of epidermal branchings of this plant, and the epidermal branching of *A. thaliana* can reduce the damage of ultraviolet light to chloroplasts and reduce internal water loss to some extent [7,8]. In wheat (*Triticum aestivum*), endoreplication in the leaf epidermal cells promotes cell expansion and accelerates cell growth [9]. When *Nicotiana benthamiana* plants are attacked by pathogens, *N. benthamiana* with a higher level of endoreplication tends to have stronger resistance [10]. In addition, studies have shown that endoreplication helps to regulate plant growth under various environmental stresses, such as high salt, drought, and low temperature among others [11].

The normal development of the maize (*Zea mays*) endosperm is critical for the yield of grain. It has been found that maize endosperm cells begin to switch from mitosis to endoreduplication during the middle and late stages of development. During this process, the cell size increases and leads to rapid grain filling and an increase in the weight of grain. This process has led to the hypothesis that the endoreduplication of endosperm cells is a key driving force for grain morphogenesis and the formation of yield [12]. Current *CKIs* in plants are members of the KIP-RELATED PROTEINS (*KRP*) family, which is homologous to the *Cip/Kip* family in animals. In general, the *KRP* family can be divided into two subfamilies, including *KRP* and *SMR*. A total of 18 *KRP* family genes have been identified in *A. thaliana* [13]. In maize, 12 members of the *SMR* family were identified. Larkins found that the expression of *Zeama; KRP;2* decreased during the development of maize endosperm [14], whereas *Zeama; KRP;1* accounted for only a portion of the observed activity of *CKI*. Moreover, *ZmSMR4* can increase the cell ploidy by inhibiting *CDKA;1/CDKA;3/CDKB1;1* and significantly promote endoreplication [15].

Elucidating the functions of key genes and the potential molecular mechanisms of endoreplication will enrich the theoretical knowledge of cell-cycle regulation and then provide a theoretical basis for the improvement of maize by regulating the mode of occurrence and process of endogenous polyploidy through molecular techniques. Based on previous laboratory studies, we found that ZmSMR10 began to be highly expressed on day 12 after pollination, which coincides with a high level of endoreplication that occurs within the maize endosperm [16]. It is important to hypothesize about the origin of endoreplication. We characterized the function of this gene by overexpression and silencing. The size and position of the protein were determined by its expression in prokaryotes and subcellular localization. The interacting proteins were further explored by yeast two-hybrid, bimolecular fluorescence complementation (BiFC), and co-immunoprecipitation (CoIP). These results led to our hypothesis. They not only provide valuable insights into the regulation of endoreplication but also provide references for the study of other genes.

## 2. Results

### 2.1. Cloning of ZmSMR10 and the Generation of Transgenic Plants

*ZmSMR10* is located on chromosome 5 of maize and produces a 15.3 kDa protein with an isoelectric point of 9.0 and no introns [17]. The overexpression of *ZmSMR10* led to transgenic *A. thaliana* plants that were smaller than the wild type (WT) until the shoot stage (Figure 1A). Correspondingly, gene-silenced maize was also higher than the control (Figure 2A,B). In addition, the leaf margins of transgenic *A. thaliana* were more distinctly serrated at the seedling stage than those of the WT (Figure 1D). This phenomenon is owing to changes in the distribution and concentration of growth hormones in the plant, which results in uneven growth division and serration of the leaf margin cells [18]. Moreover, there were more stem branches in the transgenic *A. thaliana* than in the WT (Figure 1B,C). This phenomenon is related to the different proportions and locations of hormones in the plant, such as abscisic acid (ABA) and indole acetic acid (IAA) [19]. In addition, although transgenic *A. thaliana* grew smaller than the WT, it shoots significantly earlier than the WT. Consequently, it flowers and fruits earlier (Figure 1E,F), and this early maturity trait is significant for production. Furthermore, the seeds of transgenic *A. thaliana* were fuller than those of the WT (Figure 1G,H), which is consistent with the phenotypes exhibited by other genes in the same family. We also found that mutant maize seeds were flatter, lighter, and less shiny than the WT seeds, and the length and volume of the mutant kernels were significantly smaller than those of the control. This indicated that the deletion of *ZmSMR10* severely hindered the development of kernels (Figure 2D–F). Observation of the seed sections revealed that the mutant maize endosperm was smaller and had relatively more chalky starch and little or no angular starch compared with the control (Figure 2C). Thus, the 1000-kernel weight of the mutant was significantly lower than that of the control (Figure 2G). We observed the morphology of leaf epidermal trichomes in both transgenic *A. thaliana* and its WT control because the leaf epidermal trichomes are an endoreduplication system. The results revealed that the transgenic *A. thaliana* grew a large number of bifurcated trichomes in contrast to the WT, which had more trifurcated trichomes (Figure 1I). This is similar to a previous study in which multiple clusters of multiforked trichomes appeared on the leaf surface of transgenic *A. thaliana* after the homologous *SIAMESE (SIM*) gene was silenced. This suggests that ZmSMR10 reduced the ability of *A. thaliana* trichomes to diverge by inhibiting the cell cycle.

### 2.2. Changes in the Cell Ploidy of Transgenic Plants

The ploidy levels in both the transgenic and WT *A. thaliana* at the shoot stage were determined by flow cytometry. The results indicated that in addition to the normal two, four, eight, and sixteen C cell populations, a thirty-two C cell population was observed in the transgenic *A. thaliana* as shown in Figure 3A. Furthermore, the level of cell ploidy in the transgenic *A. thaliana* was higher than that in the WT *A. thaliana* (Figure 3B) with a significantly higher C-value (Figure 3C). This indicates that the overexpression of *ZmSMR10* can increase the ploidy of plant cells. In contrast, the mutant maize had only two and four C cells (Figure 4A), and the maize ploidy was reduced after *ZmSMR*10 had been silenced (Figure 4B). Based on the cellular ploidy changes in the transgenic *A. thaliana* and mutant maize, we hypothesized that *ZmSMR10* is an important regulatory gene that can promote the phenomenon of endoreduplication (Figure 4C).

### 2.3. Effects of ZmSMR10 Gene on Seed Germination under Abscisic Acid Stress

To explore the influence of the *ZmSMR*10 gene on seed germination, we sowed transgenic *A. thaliana* plants on MS media with different concentrations of abscisic acid (ABA) and checked the rate of germination. There were high rates of germination of both the WT and transgenic *A. thaliana* in the absence of ABA. The rates of germination of both types of seeds decreased upon the addition of 1 μM ABA, which indicated that ABA can inhibit germination [20]. However, the transgenic seeds were significantly more inhibited than the WT seeds. Similarly, at ABA concentrations of 4 μM and 8 μM, the transgenic seeds germinated at a significantly lower rate than those of the WT (Figure 5A,B). Therefore, we concluded that the overexpression of *ZmSMR10* inhibited the germination rate of transgenic seeds. In addition, a similar result was observed in the maize mutant that germinated 1 day earlier than the WT (Figure 2H). This indicated that *ZmSMR10* may play a role in the molecular pathway of the regulation of seed germination by ABA.

### 2.4. Effects of ZmSMR10 Gene on the Response to Drought and Salt Stress

The promoter region of *ZmSMR10* contains ABA-responsive elements (ABRE), which suggests that it could respond to ABA and drought stress [21]. To verify this, the B73 inbred line was used to analyze the pattern of expression of *ZmSMR10* after treatment with 100 μmol L^−1^ ABA and 15% polyethylene glycol (PEG) using quantitative real-time reverse-transcription PCR (qRT-PCR). The results revealed that the *ZmSMR10* gene was significantly upregulated from 12 to 24 h and similarly increased from 12 to 24 h after the rehydration treatment (Figure 6). This also indicated that *ZmSMR10* responded to the ABA and PEG treatments. Subsequently, we verified this phenomenon by treating the plants with ABA and PEG. After 7 days of treatment, the growth and development of transgenic *A. thaliana* were inhibited by 8 µM ABA compared with the WT (Figure 7A,B). However, a 10% concentration of PEG significantly inhibited the WT *A. thaliana,* and the transgenic plants grew better (Figure 7A–C). The mutant grew much worse overall than the control after treatment with 200 μM ABA for 7 days (Figure 8A,B). After treatment with 20% PEG, the leaves of the mutant maize wilted more than the WT, which indicated that the mutant maize was more sensitive to PEG (Figure 8A–C). All these phenomena indicate that *ZmSMR*10 is sensitive to ABA and insensitive to PEG. Thus, we hypothesized that *ZmSMR10* may use the ABA pathway to confer drought resistance.

### 2.5. Prokaryotic Expression and Localization of the ZmSMR10 Protein

ZmSMR10 was verified to be a small 15–17 KDa protein by prokaryotic expression (Figure 9). Subcellular localization experiments indicated that ZmSMER10 is localized in the nucleus, which could provide insights into its function (Figure 10).

### 2.6. Interactions of the ZmSMR10 Protein

Based on the references and protein interaction prediction from the STRING website, we proposed that the three proteins ZmCDKA;1, ZmCDKA;3, and ZmCDKB1;1 interact. They are part of the cell-cycle protein kinase family. The yeast two-hybrid experiment indicated that ZmSMR10 did not interact with ZmCDKA;1 or ZmCDKA;3. It interacted only weakly with ZmCDKB1;1 in the SD-T-L-H triple-deficient medium (Figure 11A). To explore the regulatory pathway of the ZmSMR10 protein, we screened a yeast library and found that *ZmPCNA2* and *ZmCSN5B* were two very interesting genes (Figure 11B). Proliferating cell nuclear antigen2 (*PCNA2*) is a protein associated with the cell cycle that serves as an auxiliary protein for DNA polymerases. Its expression gradually increases from the G1 to the S phase and reaches a peak in the S phase [22]. The COP9 signalosome complex subunit 5b (*CSN5B*) is a protein found in *A. thaliana* that is associated with growth and development. The *CSN* gene family plays an important role in the G2 phase of the cell cycle [23].

### 2.7. BiFC and CoIP Determination of the Interaction between ZmSMR10 and ZmPCNA2, ZmCSN5B

BiFC experiments further verified the interactions between ZmSMR10 and ZmPCNA2 and ZmCSN5B (Figure 12). Finally, through CoIP experiments, Western blotting showed that ZmSMR10 strongly interacted with the ZmPCNA2 and ZmCSN5B proteins in vitro (Figure 13). Three different experiments were performed to confirm these interactions. This finding is important to elucidate the molecular mechanisms that underlie regulation of the cell cycle.

## 3. Discussion

Endoreplication plays an important role in the growth and development of maize endosperm. This has been validated in *A. thaliana*, wheat, tobacco (*Nicotiana* sp.), and maize [7,8,9,10]. *SMR* have one of the most highly conserved motifs, which contains threonine, followed by proline. This motif represents the smallest consensus site for the phosphorylation of *CDK*. This suggests that the phosphorylation of threonine may be critical for the function of *SMR*. There is also a motif that consists of a sequence in the form of PXXP, which is followed by one or more basic residues. This is similar to the structural domains involved in protein interactions, which interact with chaperone proteins by forming PP II helices [24]. After defining the *SMR* family as a class of *CKIs* by comparing homologous genes, the *SIM* gene was recognized as its first member. *SIM* was found to inhibit *CDKA;1* and the activities of *CDKB1;1*, thereby promoting endoreplication [25,26,27]. However, certain specific CKIs in maize promote the initiation of endoreplication by inhibiting m-phase CDKs [15]. Since SMR is a low-molecular-weight protein, it often does not contain introns, presumably owing to the need for rapid DNA replication and, thus, rapid cell division.

### 3.1. The ZmSMR10 Protein in Maize Is Functionally Similar to the SMR Proteins in Other Plants

The plants that overexpressed *ZmSMR*10 exhibited early shooting, flowering, fruiting, and plant dwarfing. In addition, after the *ZmSMR10* gene was silenced, the maize plants grew higher. This suggests that this gene is involved in the regulation of plant growth and development, which is consistent with previous research on *ZmSMR*4 [15] and *KRP* [28]. *ZmSMR10* also exhibited the most important function of promoting endoreduplication. Compared to the control, the plants that overexpressed this gene had a higher proportion of hyperploid cells, while the mutant plants had fewer cells with hyperploidy. This is consistent with the findings of other studies on *CKI* [15,16,17]. It is worth noting that we found that a large number of bifurcated trichomes in the leaf epidermis of transgenic *A. thaliana* overexpressing *ZmSMR*10. However, the WT *A. thaliana* has a preponderance of trifurcated trichomes. Moreover, to our knowledge, this phenomenon has not been previously documented. *A. thaliana* trichomes, a typical model organ for the study of endoreduplication, have a direct relationship between their morphology and the degree of endoreduplication. Hamdoun et al. (2016) found that *A. thaliana sim* mutants tend to show clusters and that the inclusion of *SMR1* can restore the trichome phenotype [29]. Li et al. (2019) also very intuitively illustrated that maize *ZmSMR*4 restores the abnormality of trichomes. These studies are somewhat related but not identical to our results. In addition, similar to the *ZmSMR3/4/12* genes identified to date, the *ZmSMR10* gene similarly plays an important role in the production of plump seeds [17]. Therefore, we hypothesized that this is because the *SMR* family inhibits the entire cell cycle, which causes a reduction in cell division and normal mitosis, thus resulting in a transfer to endoreplication. As a result, the chromosome of the endosperm cells multiplies in many cycles but without division, which results in the enlargement of endosperm cells and the development of fuller and larger seeds [5]. These phenotypes have appeared in previous studies on different *CKIs*, which suggested that the phenotypes may be controlled by highly conserved motifs in *CKI*. In addition, we found that the overexpression of *ZmSMR*10 resulted in a greater number of branches and a relative delay in seed germination in transgenic *A. thaliana*. Correspondingly, earlier germination was also observed in mutant maize. ABA induces the expression of *ICKs* (Cdk kinase inhibitors), which inhibit the cell cycle [30]. Combined with the phenotypes of serrated leaves, plant dwarfing, early maturity, and seed germination were observed in transgenic *A. thaliana* in this study. We hypothesized that this phenomenon may also be attributed to changes in the content and location of ABA, growth hormones, and other hormones as a result of endoreplication [19]. Changes in these hormones lead to corresponding changes in plant growth and development, which ultimately result in changes in plant morphology.

### 3.2. ZmSMR10 Is Sensitive to ABA and Resistant to PEG

We investigated whether ZmSMR10 is resistant to stress. Previously, qRT-PCR was performed, and the results showed that *ZmSMR10* responded to ABA and PEG. We found that *ZmSMR10* was sensitive to ABA and resistant to PEG after the treatment of transgenic *A. thaliana* with 8 μM ABA and 10% PEG. The same conclusion was reached through reverse validation using a mutant. This suggests that *ZmSMR10* enhances drought resistance using the ABA pathway. Grime et al. (1982) showed that the content of DNA in plants increases under conditions of low temperature [31]. Ceccarelli et al. (2006) and Vlieghe et al. (2005) reported that a high level of internal replication also improves salt [32] and drought tolerance in plants [33]. All these studies confirm that endoreplication may be a special cell cycle that plants have evolved to adapt to severe environments.

### 3.3. ZmSMR10 Interacts with ZmPCNA2/ZmCSN5B in the Nucleus

As is well known, the protein encoded by a gene must be in a specific position to fulfill its presumed function. It has been previously shown that all the SMR proteins in *A. thaliana* [34] and maize [15] have nuclear localization motifs. In this study, we observed that ZmSMR10 functions in the nucleus. Endoreplication is the increase in ploidy of cellular DNA, which is a process that occurs in the nucleus [35].

The molecular mechanism of cell-cycle regulation by the ZmSMR10 protein was explored, and during subsequent yeast two-hybrid validation, we found that none of the ZmSMR10 proteins interacted with the ZmCDKA;1/A;3 proteins except for ZmCDKB1;1, which only had weak interactions. This phenomenon is different from those of ZmSMR4 [18] and AtSIM [36], which suggests that the mode of regulation of the ZmSMR10 protein differs from those of the ZmSMR4 and AtSIM proteins. Therefore, after a yeast two-hybrid screening library, we identified two very interesting genes, *ZmPCNA2* and *ZmCSN5B*. PCNA2 is an auxiliary protein of DNA polymerase δ, which is a cell-cycle protein. The enzyme has a clamp-shaped three-dimensional structure that attaches to polymerase δ and slides on the clamped DNA template strand to ensure that the polymerase does not detach from the template strand [37]. This protein plays an important role in the replication [38], repair [39], and recombination of DNA [40]. Its expression gradually increases from the G1 to the S phase and reaches its highest level in the S phase, where it promotes cell replication [37]. In both *A. thaliana* and maize, *PCNA1/2* has been found to interact with CYCD and form a complex with CYCD–CDKA to regulate the cell cycle [41,42,43]. Another interesting gene is *COP9* signalosome complex subunit 5b (*CSN5B*). CSN is an evolutionarily conserved multi-subunit protein complex that has many nuclei [44,45,46]. It plays a role in regulating various signaling and developmental processes, such as embryogenesis, cell-cycle circadian rhythms, DNA repair, and plant responses to light and hormones [47,48]. The CSN5 subunit, one of the eight subunits of the CSN complex, is encoded by *CSN5A* and *CSN5B*, which are conserved and highly homologous. The *CSN5A* and *CSN5B* subunits are assembled in vivo to form the corresponding CSN^CSN5A^ and CSN^CSN5B^ complexes [49]. Multiple types of *csn3/csn4/csn5ab* mutants have been reported to exhibit significant G2 phase arrest, which suggests that the CSNs are important for initiating the G2 phase of the cell, thus accomplishing M phase cytokinesis [23]. The appearance of these two genes revealed a new paradigm for the regulation of the cell cycle by the ZmSMR10 protein. The molecular theory of cell-cycle regulation by the *SMR* family has also been enhanced. Although the *SMR* family is known to inhibit *CDK*, it not only regulates the cell cycle through *CDK/CYC* but also enhances the level of endoreplication by regulating other cell-cycle factors from different periods.

Three approaches were used to verify that there is a strong interaction between these proteins. They included yeast two-hybrid, BiFC, and CoIP experiments. Their functions in the cell cycle were identified as the promotion of the G1 to S phase mediated by PCNA2 [37] and initiation of the transition from G2 to the M phase by CSN5B [49]. Endoreplication is accomplished in two key steps, which include activation of the S phase and inhibition of the M phase [50]. The findings of previous studies led to a prediction of the molecular regulatory pathways by which ZmSMR10 promotes replication in plants. When a cell receives a signal indicating the need for endoreplication in the plant, highly expressed ZmSMR10 may enhance the activity of the ZmPCNA2 protein, which leads to extensive cell replication. Simultaneously, it may also inhibit the activity of the ZmCSN5B protein, which results in cells that cannot divide normally. At this point, the highly active ZmPCNA2 protein activates the G1 to S phase transition, which enables the DNA to continue replicating. In the G2 phase, the cell, which once again stagnates, undergoes a few rounds of endoreplication. The amount of DNA in the cells eventually increases exponentially. Therefore, the overexpression of *ZmSMR10* causes an increase in plant ploidy, which results in changes in plant growth and development, phenotype, and stress resistance (Figure 14).

## 4. Materials and Methods

### 4.1. Experimental Materials

The WT *A. thaliana* Columbia Zero (Col-0) and maize (*Zea mays* L.) inbred line B73 were provided by the Key Laboratory of Maize Biology and Genetic Improvement, College of Agriculture, Northwest A&F University (Xianyang, China). Maize mutant EMS4-1a6912 was purchased from the Maize Mutant Bank (EMS), Qilu Normal University (Jinan, China) (http://elabcaas.cn/memd/public/index.html#/pages/search/geneid, accessed on 7 June 2022) [51]. WT and transgenic *A. thaliana* seedlings were germinated and incubated at 4 °C on Murashige and Skoog (MS) plates for three days. They were then grown in a greenhouse at 22 °C with 70% relative humidity (RH) and a 16 h/8 h light/dark photoperiod.

### 4.2. Cloning and Transformation of ZmSMR10

The *ZmSMR10* gene sequence information was obtained from the NCBI (https://www.ncbi.nlm.nih.gov/, accessed on 10 December 2021). RNA was extracted from the endosperm 15 days after pollination using a Plant RNA Kit (Tiangen, Beijing, China). A Fast Quant RT Kit (with DNase) (Tiangen) was used to reverse-transcribe the cDNA of *ZmSMR10*. Specific primers were designed at the 5′ and 3′ ends of the mRNA using Primer-BLAST from the NCBI website, synthesized, and purified by the Beijing Aoke Dingsheng Biotechnology Co. (Beijing, China). *ZmSMR10* cDNA was obtained using the primer pair F-AGCCTCGCTTATTTAGCCTCC and R-CATCACACTCTGCTTGAGCC. The PCR mixture (50 µL) contained 5 µL of reaction buffer, 200 μM of dNTPs, 5 μM of primers, 20 ng of cDNA as a template, and 1 unit of Q5^®^ High-Fidelity DNA Polymerase (New England Biolabs, Ipswich, MA, USA). The PCR was amplified under the following conditions: initial denaturation at 98 °C for 30 s, followed by denaturation at 98 °C for 10 s, annealing at 60 °C for 30 s, elongation at 72 °C for 30 s, and final extension at 72 °C for 2 min. The amplified PCR products were gel-purified and inserted into a Pcambia1300 vector. The correctly sequenced vector was transferred into *Agrobacterium rhizogenes* GV3101 before the gene was transferred into *A. thaliana* using the inflorescence infestation method [52]. Transgenic plants were selected on plates that contained 20 µg/mL thaumatin and transplanted into the soil. The T3 generation lines were used for phenotyping.

### 4.3. Quantitative Real-Time Reverse-Transcription PCR (qRT-PCR)

Maize seedlings were cultured to the 3-leaf stage in the soil and treated with 100 μmol L^−1^ of ABA and 15% PEG. Samples were collected at 0, 2, 4, 6, 8, 10, 12, 24, 48, and 72 h after treatment. The seedlings were rehydrated after 72 h and then sampled again at 12, 24, 48, and 72 h after rehydration. All the treated and control maize materials were harvested at different time points by snap freezing in liquid nitrogen and storage at −80 °C. Total RNA and cDNA were extracted and synthesized separately, as described in Section 4.2. The pattern of expression of the *ZmSMR10* gene under ABA and PEG was analyzed by qRT-PCR (QuantStudio7 Flex, Life Technologies, Thermo Fisher Scientific, Waltham, MA, USA) using the *ZmSMR10* specific primers F-TCAAGGACCATGGCGTCAAA,R-AACGGATCCATCCTCGGGAA and the actin primers F-TGGGCCTACTGGTCTTACTACTGA and R-ACATACCCACGCTTCAGATCCT from Zm00001eb385900. All the treatments were performed in three independent biological replicates using the 2^−ΔΔCt^ method [53]. The relative values of expression were log^2^ transformed and plotted using DPS software (15.10 advanced version).

### 4.4. Plant Flow Cytometry Analysis

The sixth leaves of transgenic and control *A. thaliana* were rapidly cut into filaments and placed in 500 μL of nuclear isolation buffer (10 mmol L^−1^ MgSO_4_•7H_2_O, 50 mmol L^−1^ KCl, 5 mmol L^−1^ HEPES, 3 mmol L^−1^ DTT, and 0.2% Triton X-100), with the pH adjusted to 7.5 with NaOH to release more nuclei. The suspension was then filtered using a 50 μm filter. Next, 200 μL of the recovery solution was added, and the nuclei were stained with 4,6-diamidino-2-phenylindole (DAPI) (final concentration of 2 μg/mL) for 30 min on ice. The nuclear suspensions were analyzed by flow cytometry (CytoFLEX, Beckman Coulter, Shanghai, China). In this study, cyclic values (C-values) were used as an index to assess the degree of internal replication, and the C-values were calculated based on the weighted average of the number of nuclear ploidy in different cells. The cycling value was calculated using the following formula:C-value = [(2c × 0) + (n4c × 1) + (n8c × 2) + (n16c × 3) + (n32c × 4)]
n2c + n4c + n8c + n16c + n32c
where the C-value is the cyclic value experienced by the cell; n2c–n32c represents the number of 2c–32c ploidy nuclei particles in the nucleus, and the DNA content of a haploid nucleus is defined as 1c.

### 4.5. Subcellular Localization

The fusion expression vectors 35s: green-GFP-*ZmSMR10*, pGFP-*ZmSMR10*/35s: green-GFP, and nuclear marker were constructed by homologous recombination and transferred into the GV3101 strain by chemical transformation. The pGFP-*ZmSMR10* bacteriophage and nuclear marker bacteriophage were mixed in a 1:1 ratio as the experimental group, while the empty vector 35s: green-GFP nuclear marker bacteriophage was mixed in the same ratio as the control group. The OD of the bacterial solution was adjusted to 0.8 using a diluent of *Agrobacterium* that consisted of 10 mM morpholine ethanesulfonic acid, 50 mM MgCl_2_•6H_2_O, and 0.15 mM acetosyringone solubilized in dimethyl sulfoxide (DMSO). Subsequently, the bacterial solution was incubated in the dark for 2 to 3 h at room temperature. Finally, the bacteriophage solution was injected into 4-week-old tobacco plants and grown in a greenhouse at 22 °C at a relative humidity (RH) of 70% and a 16 h/8 h light/dark photoperiod. The samples were observed using a confocal laser scanning microscope between 48 and 72 h (Leica TCS SP8, Leica, Wetzlar, Germany).

### 4.6. Prokaryotic Expression of the ZmSMR10 Protein

The expression vector pCold-*ZmSMR10* was constructed and transferred into the ArcticExpress (DE3) pRARE2 strain (Vazyme, Dalian, China) by chemical transformation and then incubated in Luria-Bertani (LB) medium that was augmented with 100 μg/mL benzylpenicillin, 40 μg/mL gentamicin, and 25 μg/mL chloramphenicol at 37 °C until the cells reached OD = 0.8. Afterwards, 0, 2, 4, 6, and 8 mM isopropyl β-D-1-thiogalactopyranoside (IPTG) were added to the bacterial solution and divided into two groups. One group was induced at 16 °C and the other at 21 °C. The samples were collected at 0, 4, 8, 12, and 16 h. They were then mixed with SDS loading buffer after boiling in a water bath and detected using protein gels.

### 4.7. Yeast Two-Hybrid Assay

For the yeast two-hybrid assay, the expression vectors PGBKT7-*ZmSMR10* and pGADT7-*ZmCDKA1/ZmCDKA3/ZmCDKB* were constructed. We then co-transformed the experimental vectors and positive–negative control into the yeast strain Y2HGold using the lithium acetate (PEG/LiAc) method according to the manufacturer’s instructions (Puente, Wuhan, China). After 3 days, the yeast strains were dispersed in defective-Trp-Leu medium (SD-T-L), and clones were collected from the two deficient media and shaking culture. The bacterial fluids were then co-coated on both the three-deficient medium of Trp/Leu/His (SD-T-L-H) and the four-deficient medium of Trp/Leu/His/Ade (SD-T-L-H-A) to conduct interaction tests. The expression activity of the reporter gene LacZ was characterized using a β-galactosidase assay to determine whether the plaque changed blue.

### 4.8. Yeast Two-Hybrid Screen Library

First, PGBKT7-*ZmSMR10* and pGADT7 were co-transfected into Y2Hgold for self-activation validation, and PGBKT7-*ZmSMR10* was then transferred into Y2Hgold and co-cultured with yeast two-hybrid libraries at 28 °C for 8 h using the mating method according to the manufacturer’s instructions (Ouyi Bio, Shanghai, China). After the Mickey head was observed under the microscope, the bacterial solution was coated on defective-Trp/Leu/His/Ade (SD-T-L-H-A) media and incubated at 28 °C for 3 days. The grown plaques were collected in SD-T-L-H-A media that contained X-α-Gal for 3 days at 28 °C for the second validation, and the plaques were sequenced. Finally, the reciprocal genes that were obtained were subjected to yeast two-hybrid reversal to verify their interactions, as described in Section 4.7.

### 4.9. Bimolecular Fluorescence Complementation (BiFC)

We constructed the expression vectors 35S-SPYNE-*ZmSMR10* and 35S-SPYCE-Zm*PCNA/ZmCSN5B* and transferred the correctly sequenced plasmids into *Agrobacterium rhizogenes* strain GV3101 using the chemical transformation method. Yellow fluorescence was observed under a confocal microscope after injecting the experimental and control groups into *N. benthamiana*, as described in Section 4.5.

### 4.10. Co-Immunoprecipitation (CoIP)

The expression vectors pSuper1300-GFP-*ZmSMR10* and pCXSN-HA-*ZmPCNA/ZmCSN5B* were constructed to express the genes in *N. benthamiana*, as described in Section 4.5, and liquid nitrogen samples were collected and stored at −80 °C after 48 h. The protocol provided by the IP kit (Lablead, Beijing, China) was used for the protein extraction and IP experiments. Western blotting [54] was used to identify interactions between the genes.

## 5. Conclusions

In this study, we found that *ZmSMR*10 promoted endoreplication and had many beneficial functions, including the production of fuller seeds and the induction of early maturity, multiple branching, and plant dwarfing, which provides valuable information for crop breeding. It was also verified that high levels of endoreplication affect the plant morphology, germination time, and drought resistance by altering hormones in vivo. ZmSMR10 promotes endoreplication by interacting with ZmPCNA2 and ZmCSN5B, which is highly significant for breeding to improve the molecular mechanism of plant cell-cycle regulation.

## Figures and Tables

**Figure 1 ijms-25-03356-f001:**
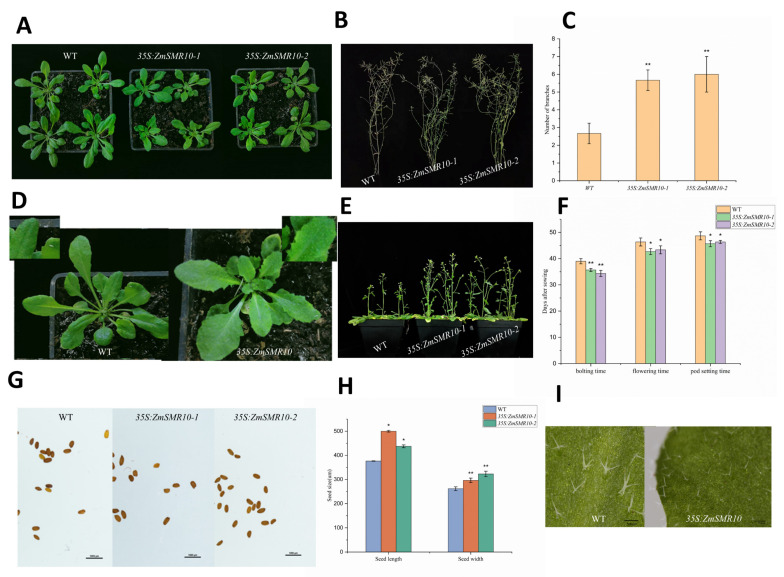
A phenotypic analysis was conducted on transgenic *Arabidopsis thaliana* lines transduced with 35S: ZmSMR10. The following comparisons were made between the wild-type (WT) and transgenic *A. thaliana*: (**A**) plant size; (**B**) number of branches at maturity; (**C**) number of branches at maturity for the WT and transgenic *A. thaliana*; (**D**) leaf margin serration; (**E**) speed of shoot extraction and flowering; (**F**) days to shoot, flowering, and fruiting for the WT and transgenic *A. thaliana*; (**G**) seed-size fullness under microscope (Bar = 1000 μM); (**H**) seed length and width values for the WT and transgenic *A. thaliana*; and (**I**) leaf epidermal trichome morphology (Bar = 500 μM). The order of description above corresponds to left to right in the figure. Independent *t*-tests showed significant differences between the WT and transgenic *A. thaliana* (** *p* < 0.01) for (**C**), (* *p* < 0.05) for (**F**) days to flowering and fruiting, highly significant differences (** *p* < 0.01) for (**F**) days to shoot, and significant differences (* *p* < 0.05) for (**H**) seed length and highly significant differences (** *p* < 0.01) for (**H**) seed width.

**Figure 2 ijms-25-03356-f002:**
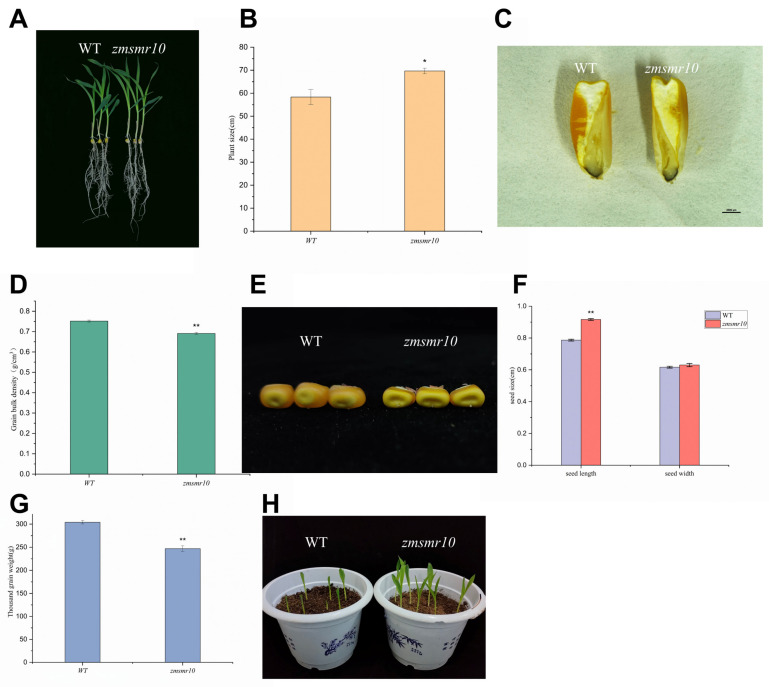
Maize mutant phenotypes resulting from the EMS mutation *zmsmr10*. (**A**) At the 3-leaf-1-heart stage, the *zmsmr10* plants were significantly smaller than the WT plants. (**B**) There was a significant difference in height between the two groups at the 3-leaf-1-center stage (**C**) The cross-section structure of the seeds under the microscope was observed. (**D**) There was a highly significant difference in grain capacity between the two groups. (**E**) The seed fullness was also compared between the two groups. (**F**) The length and width values of the seeds were detected, and it was found that there was a highly significant difference in seed length between the two groups** but no significant difference in seed width. (**G**) The 1000-grain weight statistics of the two groups were compared, and it was found that there was a highly significant difference in the 1000-grain weight between the two groups. (**H**) The *zmsmr10* plants germinated one day later than the WT plants. These experiments were repeated three times with > 100 seeds in each trial. All the data presented are the means ± SE from three independent experiments. * *p* < 0.05. ** *p* < 0.01.

**Figure 3 ijms-25-03356-f003:**
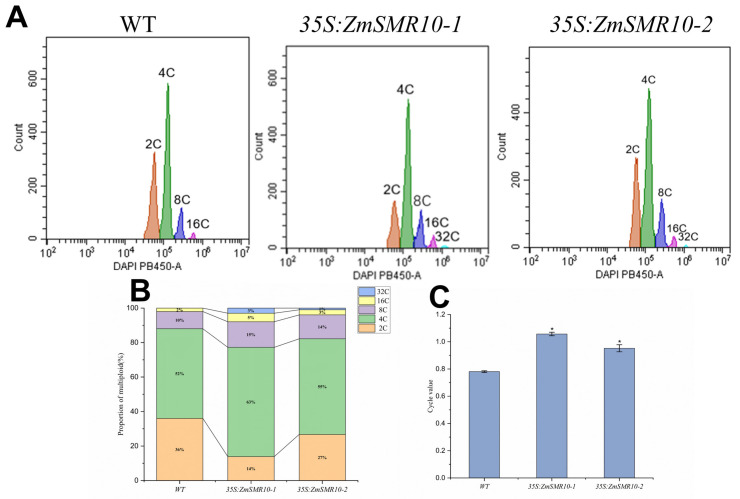
Cellular ploidy analysis was conducted on *Arabidopsis thaliana* transgenic lines that were transduced with 35S: ZmSMR10. (**A**) The DNA ploidy assay of leaves from both the WT and transgenic *A. thaliana* was examined at the shoot stage. (**B**) The ratio of cells with different karyotypes in the leaves of both WT and transgenic *A. thaliana* at the shoot stage was also evaluated. (**C**) The intracellular replication indices in leaves of the WT and transgenic *A. thaliana* at the shoot stage were analyzed, and significant differences in the C-values between the WT and transgenic *A. thaliana* were observed. The means ± SE (bar) were obtained from three independent experiments. * *p* < 0.05.

**Figure 4 ijms-25-03356-f004:**
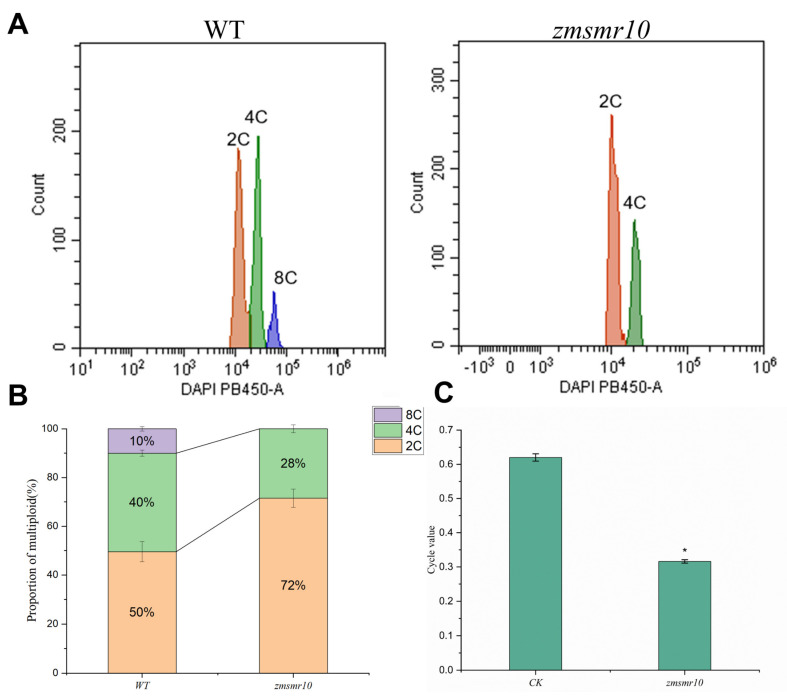
A cellular ploidy analysis was conducted on the maize mutants with the EMS mutation *zmsmr10*. (**A**) A DNA ploidy assay was performed on the leaves of both the WT and *zmsmr10* at the 3-leaf-1-heart stage. (**B**) The ratio of cells with different karyotypes was evaluated in leaves of both the WT and *zmsmr10* at the three-leaf monocarp stage. (**C**) Statistical analysis of intracellular replication indices was also carried out on leaves of both the WT and *zmsmr10* at the three-leaf monocarp stage. The mean ± SE were calculated from three independent experiments. Independent *t*-tests demonstrated significant differences in the C-values between the WT and *zmsmr10*. Ethylmethanesulfonate, EMS; WT, wild-type. * *p* < 0.05.

**Figure 5 ijms-25-03356-f005:**
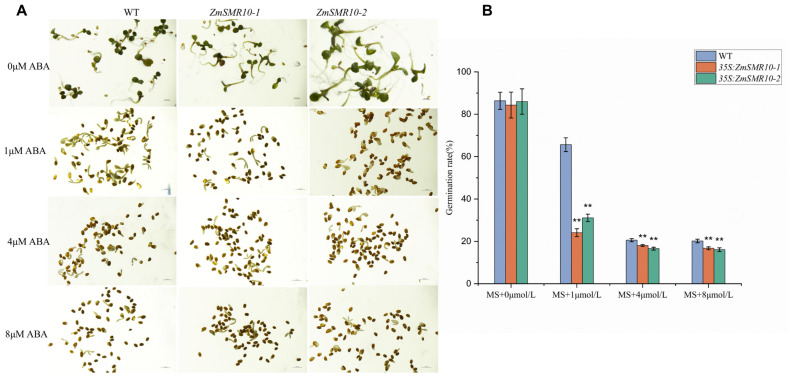
The germination rates of *Arabidopsis thaliana* transgenic lines that were transduced with 35S: ZmSMR10. (**A**) The seeds of both the wild type (WT) and transgenic *A. thaliana* were grown under different concentrations of ABA. The ABA concentrations used were 0 μM, 1 μM, 4 μM, and 8 μM ABA. (Bar = 1000 μM) (**B**) The germination rates between the WT and lines 1 and 2. (n = 3), and data are the mean ± SE (bar). ABA, abscisic acid. ** *p* < 0.01 (*t*-test).

**Figure 6 ijms-25-03356-f006:**
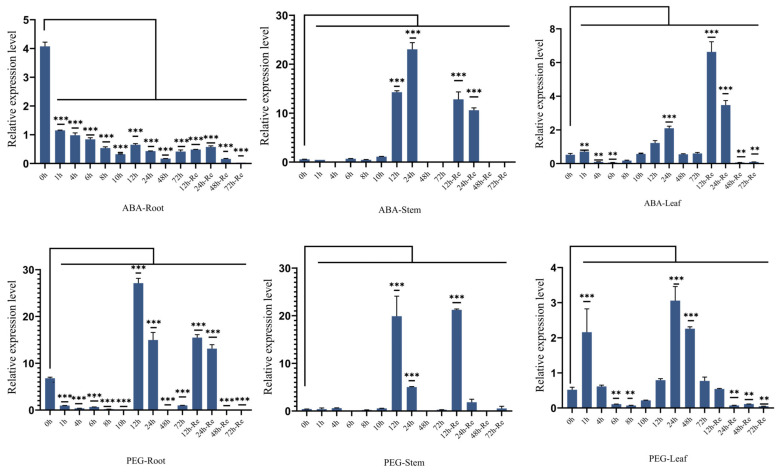
qRT-PCR quantification of *ZmSMR10* in maize leaves, stems, and roots of B73 treated with 100 μM ABA and 15% PEG. The horizontal coordinates are 0 h, 1 h, …, 72 h, the time after ABA and PEG treatments, 12 h-Re…7 2h-Re, and time of re-watering after 72 h of abiotic treatment. The vertical coordinate is the relative expression of *ZmSMR10*. Data are the mean ± SE (bar). n = 3. ** *p* < 0.01, *** *p* < 0.001.

**Figure 7 ijms-25-03356-f007:**
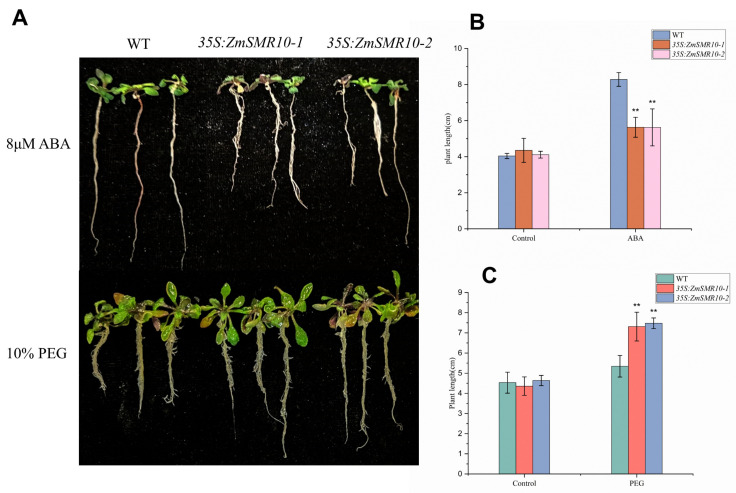
Trans-35S: *ZmSMR10 Arabisopsis thaliana* lines subjected to reverse analysis under 8 μM ABA and 10% PEG treatments. (**A**) The plant size of WT, 35S: *ZmSMR10-1*, and 35S: *ZmSMR10-2* were measured after the treatment with 8 μM ABA (**top**) and 10% PEG (**bottom**). (**B**) The plant length statistics of WT, 35S: *ZmSMR10-1*, and 35S: *ZmSMR10-2* were also measured after the treatment with 8 μM ABA. (**C**) The plant length statistics of WT, 35S: *ZmSMR10-1*, and 35S: *ZmSMR10-2* were also measured after the treatment with 10% PEG. ** Means and standard errors (bar) were calculated from three independent experiments. ABA, abscisic acid; PEG, polyethylene glycol; WT, wild type. ** *p* < 0.01. (Independent *t*-tests. n = 3.).

**Figure 8 ijms-25-03356-f008:**
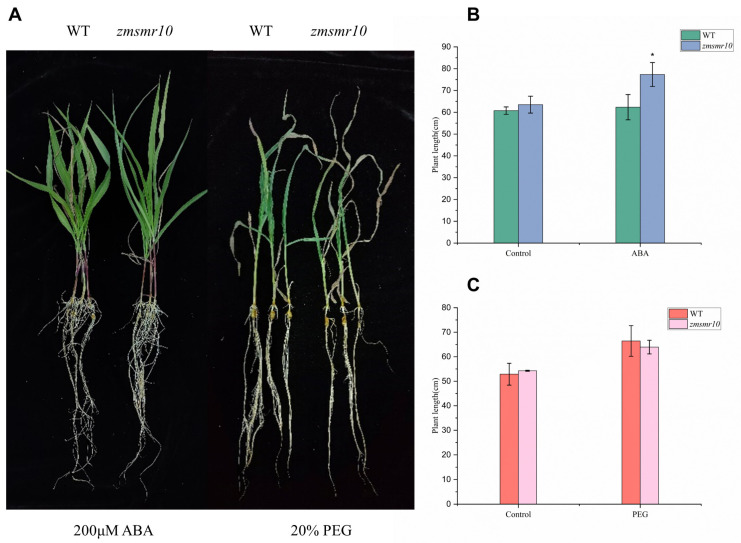
Adverse analysis of maize mutants with the EMS mutation *zmsmr10* under 200 μM ABA and 20% PEG treatments. (**A**) Size of WT and *zmsmr10* after 200 μM ABA (**left**) and 20% PEG (**right**). (**B**) Length of WT and *zmsmr10* plants after 200 μM ABA. (**C**) WT and *zmsmr10* after 20% PEG treatment. *zmsmr10* had more severe leaf desiccation. Data are the mean ± SE (bar). ABA, abscisic acid; PEG, polyethylene glycol; WT, wild type. * *p* > 0.05. n = 3.

**Figure 9 ijms-25-03356-f009:**
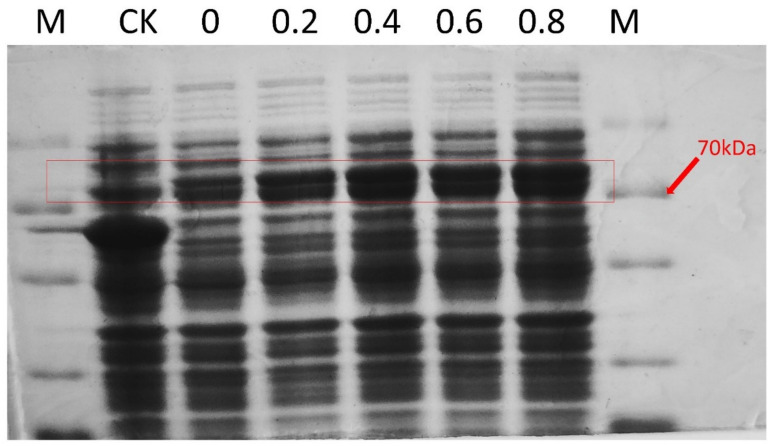
ZmSMR10-pCold fusion protein (ZmSMR10-pCold) was induced at 0.2/0.4/0.6/0.8 mM IPTG at 16 °C. M, protein marker. The target protein, which is approximately 70 kDa in size, can be seen in the red box in the figure.

**Figure 10 ijms-25-03356-f010:**
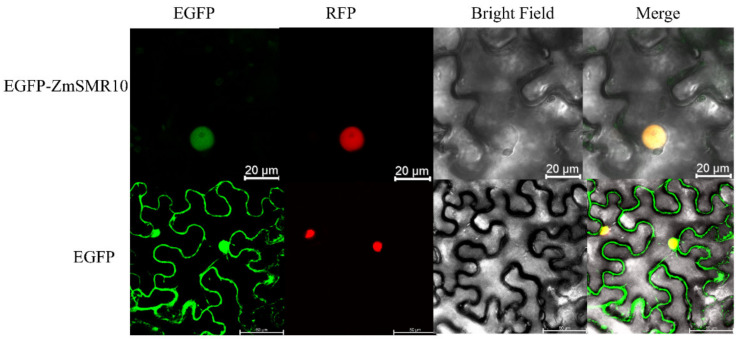
Subcellular localization of ZmSMR10. From left to right: eGFP (target protein), RFP (nuclear marker), bright field, and mixed field. The top row is the EGFP-ZmSMR10 fusion protein (bar = 20 μm) and the bottom row is the GFP-empty protein as a control (bar = 50 μm). eGFP, enhanced green fluorescent protein; RFP, red fluorescent protein.

**Figure 11 ijms-25-03356-f011:**
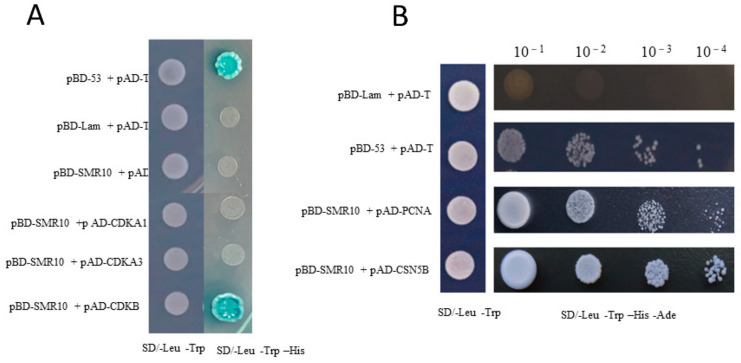
A plot of the ZmSMR10 protein interaction analysis in yeast. (**A**) The interaction of ZmSMR10 protein with ZmCDKA;1/ZmCDKA;3/ZmCDKB was verified by a yeast two-hybrid study. The top-to-bottom order of the experimental groups is as follows: positive control, negative control, self-activation validation, and three pairs of experimental groups. Left, SD/-Leu-Trp solid medium; right, SD/-Leu-Trp-His solid medium coated with x-α-gal. (**B**) The plot shows that ZmPCNA2/ZmCSN5B were screened by the yeast sieve library assay and further verified by a yeast two-hybrid study for interactions between the ZmPCNA2/ZmCSN5B and ZmSMR10 proteins. The SD/-Leu-Trp solid medium is shown on the left, and the right is the SD/-Leu-Trp-His-Ade solid medium. The bacterial dilution concentrations are represented as 10^−1^, 10^−2^, 10^−3^, and 10^−4^.

**Figure 12 ijms-25-03356-f012:**
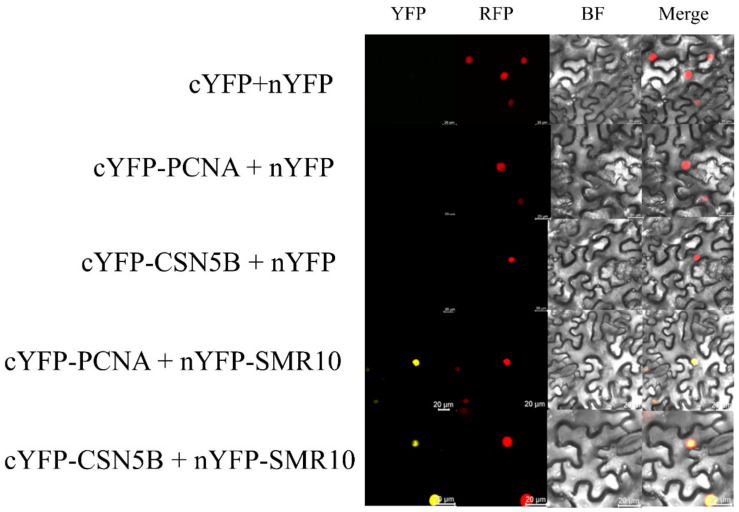
Confirmation of the interaction between ZmSMR10 and ZmPCNA2/ZmCSN5B using BIFC experiments. Red, nuclear marker. BIFC, bimolecular fluorescence complementation; YFP, yellow fluorescence protein (Bar = 20 μm).

**Figure 13 ijms-25-03356-f013:**
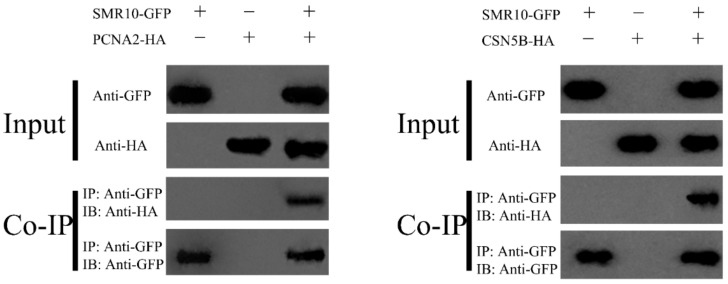
Verification of the interactions of ZmSMR10 with ZmPCNA2/ZmCSN5B in vivo by a CoIP assay. CoIP, co-immunoprecipitation; GFP, green fluorescent protein; HA, hemagglutinan.

**Figure 14 ijms-25-03356-f014:**
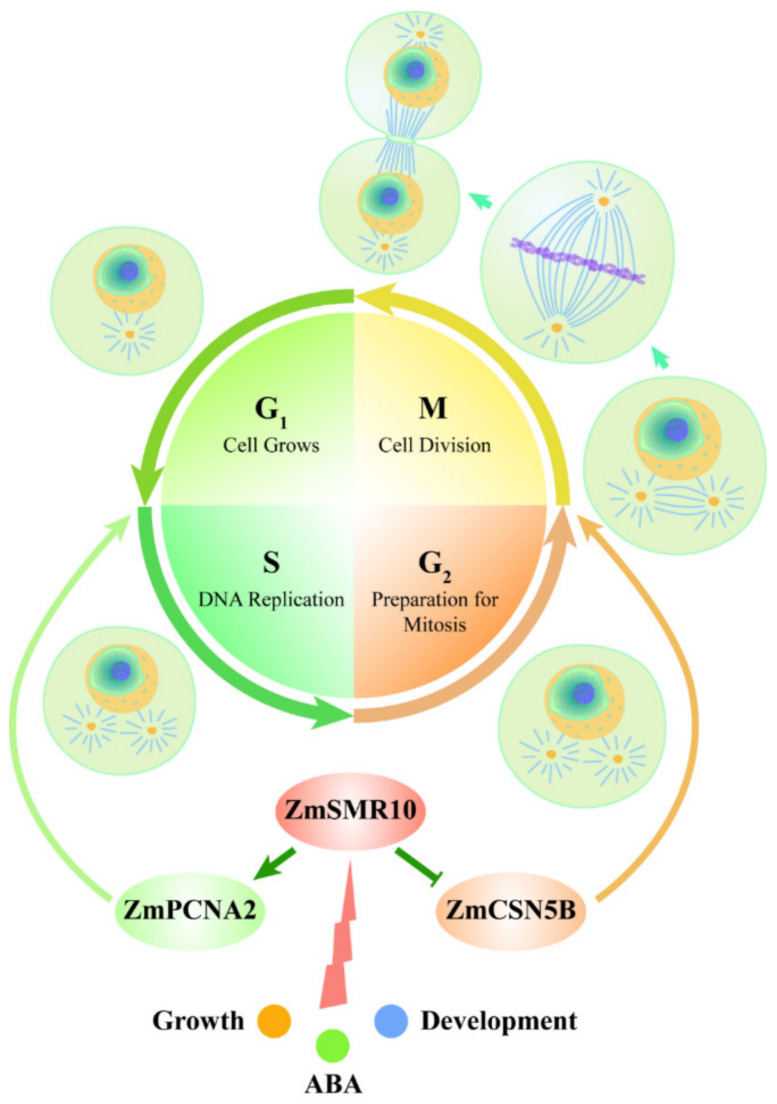
Molecular regulatory mechanism of the ZmSMR10 protein.

## Data Availability

The data presented in this study are available on request from the corresponding author. The data are not publicly available due to privacy or ethical concerns.

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
