# Peer review of "ZmSMR10* Increases the Level of Endoreplication of Plants through Its Interactions with *ZmPCNA2* and *ZmCSN5B"

_ijms, 2024, doi:10.3390/ijms25063356_

Round 1

Reviewer 1 Report

Comments and Suggestions for Authors There are several areas in which the manuscript could be improved to enhance the originality of the results and provide a more comprehensive understanding of the research.

1.       Line20-22Seems confusing, please rephrase this part.

2.       The author should provide a reasonable explanation about why not overexpressing ZmSMR10 in maize.

3.       Please double check the genes and proteins's name. Genes must be in Italic but not proteins. In addition, Arabidopsis thaliana must be in Italic but not Arabidopsis.

4.       Line 89-90:growth hormones and other hormones”. Seems confusing, please rephrase this part.

5.       The lack of functional analyses is the main drawback of this work. The author should provide references with direct functional proof to supporting that ZmSMR10, ZmPCNA2 and ZmCSN5B involved in cell cycle regulation.

6.       The author provide detected the hormones content in transgenic plants.

7.       Figures: The figure font size should be uniform. Figure 1I lack scale bar. In addition, the statistical analysis is missing in figure 6.

8.       Line 185: 200mM ABA????

9.       Line 210-212: Seems confusing, please rephrase.

10.   Line 219: EGFP-ZmSMR10 proteome??

11.   Line 244: ZmSMR10, ZmPCNA2 and ZmCSN5B.

12.   Line 247: CoIP.

13.   Figure 13: The author should use GFP+PCNA2-HA/CSN5B-HA as negative control.

14.   Line 281: Furthermore, this phenomenon has not been previously reported. ????

15.   Line 351-352: It is not clear how the interaction between them supports the role of them in cell cycle.

16.References format in the text need revision. Comments on the Quality of English Language The manuscript should go through substantial editing to fit the journal English standard.

Author Response

Dear Reviewer:

Thank you for your comments concerning our manuscript entitled “Paper Title” (ID: ijms-2897707). Those comments are all valuable and very helpful for revising and improving our paper, as well as the important guiding significance to our researches. We have studied comments carefully and have made correction which we hope meet with approval. Revised portion are marked in red in the paper. English has been edited and certified by a professional organisation. The main corrections in the paper and the responds to the your comments are as flowing:

  1. Response to comment: Line20-22:Seems confusing, please rephrase this part.

Response: We are very sorry for our incorrect writing and have re-written this part according to the Reviewer’s suggestion. “Unlike ZmSMR4, yeast two-hybrid experiments demonstrated that ZmSMR10 does not interact strongly with ZmCDKA;1/ZmCDKA;3/ZmCDKB1;1. Instead, it interacts strongly with ZmPCNA2 and ZmCSN5B.

  1. Response to comment: The author should provide a reasonable explanation about why not overexpressing ZmSMR10 in maize.

Response: We have re-written this part according to the Reviewer’s suggestion. “The normal development of the maize (Zea mays) endosperm is critical for the yield of grain. It has been found that maize endosperm cells begin to switch from mitosis to endoreduplication during the middle and late stages of development. During this process, the cell size increases and leads to rapid grain filling and an increase in the weight of grain. This process has led to the hypothesis that the endoreduplication of endosperm cells is a key driving force for grain morphogenesis and the formation of yield” in line 55-60.

  1. Response to comment: Please double check the genes and proteins's name. Genes must be in Italic but not proteins. In addition, Arabidopsis thaliana must be in Italic but not Arabidopsis.

Response: We are very sorry for our negligence of font italic. We have made correction according to the Reviewer’s comments.

  1. Response to comment: Line 89-90: “growth hormones and other hormones”. Seems confusing, please rephrase this part.

Response: We have re-written this part according to the Reviewer’s suggestion. “This phenomenon is related to the different proportions and locations of hormones in the plant, such as abscisic acid (ABA) and indole acetic acid(IAA).” in line 93-94.

  1. Response to comment: The lack of functional analyses is the main drawback of this work. The author should provide references with direct functional proof to supporting that ZmSMR10, ZmPCNA2 and ZmCSN5B involved in cell cycle regulation.

Response: We have re-written this part according to the Reviewer’s suggestion. This is direct evidence that ZmSMR10 inhibits the cell cycle and promotes internal replication in line 290-293.“ZmSMR10 also exhibited the most important function of promoting endoreduplication. Compared to the control, the plants that overexpressed this gene had a higher proportion of hyperploid cells, while the mutant plants had fewer cells with hyperploidy. This is consistent with the findings of other studies on CKI [15-17].”;This is direct functional evidence of the involvement of ZmPCNA2 in cell cycle regulation literature in line 349-352“Its expression gradually increases from the G1 to the S phase and reaches its highest level in the S phase where it promotes cell replication [39]. In both A. thaliana and maize, PCNA1/2 has been found to interact with CYCD and form a complex with CYCD-CDKA to regulate the cell cycle[43-45]”;This is direct functional evidence of the involvement of ZmCSN5B in cell cycle regulation literature in line 360-362.“Multiple types of csn3/csn4/csn5ab mutants have been reported to exhibit significant G2 phase arrest, which suggests that the CSNs are important for initiating the G2 phase of the cell, thus, accomplishing M phase cytokinesis [23]”

  1. Response to comment: The author provide detected the hormones content in transgenic plants.

Response: We think the reviewers' comments are excellent. We found that ZmSMR10 is closely related to plant growth and development during our experiments, and these phenomena can not be separated from hormone regulation, and the seed germination experiments also fully indicate that ZmSMR10 has a strong relationship with ABA. Therefore, we have carried out independent experiments to investigate this aspect comprehensively and in depth, and the results of this part will appear in another paper.

  1. Response to comment: The figure font size should be uniform. Figure 1I lack scale bar. In addition,the statistical analysis is missing in figure 6.

Response: We are very sorry for our incorrect writing. We have made changes to this section.

  1. Response to comment: Line 185: 200mM ABA????

Response: We are very sorry that it was an oversight on our part. We have changed the price from 200mM ABA to “200μM ABA” in line 194.

  1. Response to comment: Seems confusing, please rephrase in line 210-212.

Response: We have re-written this part according to the Reviewer’s suggestion. “ZmSMR10 was verified to be a small 15-17 KDa protein by prokaryotic expression” in line 219.

  1. Response to comment:  Line 219: EGFP-ZmSMR10 proteome??

Response: We are very sorry for our incorrect writing. We have changed it to “EGFP-ZmSMR10 fusion protein” in line 226-227

  1. Response to comment:   Line 244: ZmSMR10, ZmPCNA2 and ZmCSN5B.

Response: We are very sorry for our incorrect writing. We've changed the font in line 253-254.

  1. Response to comment:   Line 247: CoIP.

Response: We are very sorry for our incorrect writing. We've changed it to CoIP in line 256.

  1. Response to comment:   Figure 13: The author should use GFP+PCNA2-HA/CSN5B-HA as negative control.

Response: We think the reviewers' comments are excellent, which is better than PCNA2-HA/CSN5B-HA as negative control. It's just that the literature referenced during previous experiments used this method, and PCNA2-HA/CSN5B-HA as negative control is also possible. [1] Qingmei Guan, Xiule Yue, Haitao Zeng, Jianhua Zhu, The protein phosphatase RCF2 and its interacting partner NAC019 are critical for heat stress–responsive gene regulation and thermotolerance in Arabidopsis. The Plant Cell, 2014, 26, 438–453. [2] Wang G. Receptor-like cytoplasmic kinase 185 mediated chitin perception induced MAPK cascade immune signaling in rice in Chinese[D]. East China Normal University,2018. [3] Aung K, Kim P, Li Z, Joe A, Kvitko B, Alfano JR, He SY. Pathogenic Bacteria Target Plant Plasmodesmata to Colonize and Invade Surrounding Tissues. Plant Cell, 2020, 32, 595-611. But we will take this suggestion in future experiments

  1. Response to comment:   Line 281: Furthermore, this phenomenon has not been previously reported. ????

Response: We apologise for the incorrect representation. We have re-written this part to “Moreover, to our knowledge, this phenomenon has not been previously documented.” In line 296-297. The literature we have reviewed shows that CKI is first mutated away and Arabidopsis trichomes appear as multiclustered trichomes, and then CKI is overexpressed and the trichome phenotype is restored to the normal trichome phenotype. However, we have not reviewed the literature on the direct overexpression of CKI to observe the trichome state phenotype.

  1. Response to comment:  Line 351-352: It is not clear how the interaction between them supports the role of them in cell cycle.

Response: Yes, at present, we hypothesised the mode of interaction between ZmPCNA2 and ZmCSN5B based on their functions and the phenotype of ZmSMR10 in promoting endoreplication in the existing literature. In subsequent studies, we will dig more into the molecular regulation of maize endoreplication by ZmPCNA2 and ZmCSN5B and the mode of interaction with ZmSMR10.

  1. Response to comment:References format in the text need revision.

Response: We are very sorry for our negligence about references format. We've made changes to this section

We tried our best to improve the manuscript and made some changes in the manuscript. These changes will not influence the content and framework of the paper. We marked in red in “manuscript edited” word.

We appreciate for your warm work earnestly, and hope that the correction will meet with approval.

Once again, thank you very much for your comments and suggestions.

Reviewer 2 Report

Comments and Suggestions for Authors

The aim of the study was to determine the function of ZmSMR10 (Zm00001eb231280) gene that was characterized by overexpression and silencing. Compared with the control, the transgenic plants showed the characters of early maturing, dwarfing and drought resistance. Prokaryotic protein expression experiments demonstrated that ZmSMR10 is a small protein, and subcellular localization results suggested that it travels functionally in the nucleus. The Authors reported that protein interaction analysis showed that ZmSMR10 had no strong interaction with ZmCDKA;1/ZmCDKA;3/ZmCDKB1;1, but had strong interaction with ZmPCNA2 and ZmCSN5B. Hence, ZmSMR10 protein may be involved in the regulation of endoreplication through the interaction of ZmPCNA2 and ZmCSN5B.

The paper is quite interesting in the research field, however, I suggest the following improvements:

-            Figure 1 C and F; Figure 2 B, D, F ang G, Fig. 5B, Fig. 6 - the resolution should be improved,

-            There is the lack of the legend and description of the scheme on page 15,

-            Fig. 6 – there is the lack of statistical results marked over the bars,

-            Methodology of qRT-PCR quantification of ZmSMR10 gene in maize should be described more precisely, e.g. there is no information about the reference gene and its GenBank accession no.

-            Moderate editing of English language by the native speaker, specialist in the field is required.

Comments on the Quality of English Language

 Moderate editing of English language by the native speaker, specialist in the field is required.

Author Response

Dear Reviewer:

Thank you for your comments concerning our manuscript entitled “Paper Title” (ID: ijms-2897707). Those comments are all valuable and very helpful for revising and improving our paper, as well as the important guiding significance to our researches. We have studied comments carefully and have made correction which we hope meet with approval. Revised portion are marked in red in the paper. English has been edited and certified by a professional organisation. The main corrections in the paper and the responds to the your comments are as flowing:

  1. Response to comment: Figure 1 C and F; Figure 2 B, D, F ang G, Fig. 5B, Fig. 6 - the resolution should be improved.

Response: Thank you very much for your suggestion, we have made the change from figure 1 to figure 8.

  1. Response to comment: There is the lack of the legend and description of the scheme on page 15

Response: We are very sorry for our oversight. There is an error in the position of the picture and legend, we have made adjustments.

  1. Response to comment: Fig. 6 – there is the lack of statistical results marked over the bars

Response: Thank you for your suggestions, we have made additions.

  1. Response to comment: Methodology of qRT-PCR quantification of ZmSMR10 gene in maize should be described more precisely, e.g. there is no information about the reference gene and its GenBank accession no.

Response: Thank you very much for your comments, we have changed it to “The pattern of expression of the ZmSMR10 gene under ABA and PEG was analyzed by qRT-PCR (QuantStudio7 Flex, Life Technologies, Thermo Fisher Scientific, Waltham, MA, USA) using the ZmSMR10 specific primers F-TCAAGGACCATGGCGTCAAA,R-AACGGATCCATCCTCGGGAA and the actin primers F- TGGGCCTACTGGTCTTACTACTGA, R- ACATACCCACGCTTCAGATCCT from Zm00001eb385900.” in line 432-437.

  1. Response to comment: Moderate editing of English language by the native speaker, specialist in the field is required.

Response: Thank you for your advice. We have reworked it through a team of professional English language editors. The editing certificate is also in this resubmission.

We tried our best to improve the manuscript and made some changes in the manuscript. These changes will not influence the content and framework of the paper. We marked in red in “manuscript edited” word.

We appreciate for your warm work earnestly, and hope that the correction will meet with approval.

Once again, thank you very much for your comments and suggestions.

Round 2

Reviewer 1 Report

Comments and Suggestions for Authors

The authors have answered to most comments in an appropriate manner.

Comments on the Quality of English Language

Moderate editing of English language required.